# Molecular In-Depth Characterization of Chondrosarcoma for Current and Future Targeted Therapies

**DOI:** 10.3390/cancers15092556

**Published:** 2023-04-29

**Authors:** Sebastian Gottfried Walter, Peter Knöll, Peer Eysel, Alexander Quaas, Christopher Gaisendrees, Robert Nißler, Lena Hieggelke

**Affiliations:** 1Department for Orthopedic Surgery and Traumatology, University Hospital Cologne, Joseph-Stelzmann-Str. 24, 50931 Cologne, Germanypeer.eysel@uk-koeln.de (P.E.); 2Department for Pathology, University Hospital Cologne, Kerpener Str. 62, 50937 Cologne, Germany; 3Department for Cardiothoracic Surgery, University Hospital Cologne, Kerpener Str. 62, 50937 Cologne, Germany; 4Nanoparticle Systems Engineering Laboratory, Institute of Energy and Process Engineering (IEPE), Department of Mechanical and Process Engineering (D-MAVT), ETH Zurich, Sonneggstrasse 3, 8092 Zurich, Switzerland

**Keywords:** sarcoma, chondrosarcoma, immune, tumor microenvironment, extracellular matrix, tumor profile

## Abstract

**Simple Summary:**

Chondrosarcoma are rare bone tumors. So far, the treatment of choice is complete resection. In cases that cannot be resected, therapeutic options are limited. Chondrosarcoma are still poorly understood compared to other types of tumors. Characterization of specific molecules and tumor cells of chondrosarcoma will help to develop better therapies in the future.

**Abstract:**

Chondrosarcoma (CHS) are heterogenous, but as a whole, represent the second most common primary malignant bone tumor entity. Although knowledge on tumor biology has grown exponentially during the past few decades, surgical resection remains the gold standard for the treatment of these tumors, while radiation and differentiated chemotherapy do not result in sufficient cancer control. An in-depth molecular characterization of CHS reveals significant differences compared to tumors of epithelial origin. Genetically, CHS are heterogenous, but there is no characteristic mutation defining CHS, and yet, IDH1 and IDH2 mutations are frequent. Hypovascularization, extracellular matrix composition of collagen, proteoglycans, and hyaluronan create a mechanical barrier for tumor suppressive immune cells. Comparatively low proliferation rates, MDR-1 expression and an acidic tumor microenvironment further limit therapeutic options in CHS. Future advances in CHS therapy depend on the further characterization of CHS, especially the tumor immune microenvironment, for improved and better targeted therapies.

## 1. Introduction

Primary malignant bone tumors are a rare entity and thus account for less than 1% of cancers [1]. Chondrosarcoma (CHS) represent the second most common primary malignant bone tumor entity [2,3]. Although there is an overall improvement in the combined treatment of other types of bone sarcomas, there was no significant progress in the treatment of CHS within the past 40 years. Thus, surgical treatment is still the only way to cure the majority of patients suffering from CHS. CHS are heterogeneous with respect to diverse subtypes that are heterogeneous in terms of clinical and prognostic features, as well as molecular characteristics [4,5]. CHS are classified according to their location in relation to the bone. Most central CHS arise in bones formed by endochondral ossification (more than 85%), whereas the less common peripheral and periosteal CHS arise in flat bones (most commonly the scapula and pelvic bones) in a pre-existing osteochondroma. In clinical practice, CHS are still classified into three grades based on their histology, ranging from well-differentiated, low-cellularity, exceptionally metastatic grade I CHS to poorly differentiated and highly cellular grade III CHS, with a high risk for pulmonary metastasis [6,7,8]. With respect to the latest WHO recommendations, nomenclature for CHS differentiates between central or secondary peripheral atypical cartilaginous tumor (ACT)/CHS (grade 1) and central or secondary peripheral CHS grade 2 and 3 [7]. At initial presentation, only about 6% of CHS patients are diagnosed with distant metastasis [9,10]. A total of 10–30% of the patients will develop distant metastasis after local recurrence or as primary progression [11,12]. Nevertheless, an overall five-year survival rate is estimated to be around 70%, which is superior to other primary bone tumors such as osteosarcoma [3]. The survival rate of CHS depends on subtypes, with good five-year survival rates in the periosteal subtype (68.1%) and conventional CHS (68.4%). The one- and five-year survival rates in clear cell (88.7%, 62.3%), myxoid (86.2%, 49.8%), and mesenchymal (76.1%, 37.6%) subtypes are relatively lower but still better than in dedifferentiated CHS, with reported five-year survival rates between 11% and 24% [13]. This may appear contradictory as the gold-standard for the treatment of CHS is surgical resection, and neither radiation nor chemotherapy have proven to have a relevant effect on recurrence rate and prognosis. 

Research on CHS is diminutive compared to research on osteosarcoma with regard to publications per year, but it is active with regard to the characterization of the (novel) molecular markers of CHS subtypes and the identification of potential targets for immune and targeted therapies. 

## 2. Morphology and Subtypes of CHS

From a macroscopic perspective, CHS are rather large tumors, usually extending 4 cm in diameter. Typically, they present a translucent lobular, blue-grayish, or white surface associated to the presence of hyaline cartilage [14]. Microscopically, the differentiation between low-grade chondrosarcoma and an osteochondroma can be challenging, due to the fact that binucleated cells, cystic changes, and necrosis can be seen in cartilaginous cap and osteochondroma. The synopsis of clinical and radiological features together with the macroscopic appearance of a cartilaginous cap > 2 cm are essential to establish the diagnosis in the case of secondary peripheral ACT/CHS. In general, an increasing lobulation separated by fibrous bands containing small vessels into nodules of varying sizes is suspicious for the presence of an ACT/CHS1 [7]. The following features are typical for CHS: 

Grade 1: Low to moderate cellularity with embedded chondrocytes that partially show small, dense, and binucleated nuclei that are usually not enlarged. Mitotic figures are absent. The stroma is generally composed of a majority of cartilaginous tissue; myxoid areas are sparse or absent [15].

Grade 2: There is an increase in cell density, with a subsequent smaller proportion of chondroid matrix and a relatively higher percentage of myxoid stroma. The cell nuclei are of moderate size. The mitotic rate is low (<2/10 HPF, high power fields). Some cell nuclei appear enlarged, vesicular, or hyperchromatic. The majority of chondrocytes present themselves as binucleated or multinucleated [16]. 

Grade 3: In these tumors, cellularity is highest and the chondroid matrix is very scant and dominated by myxoid areas. Chondrocytes appear as irregular with a tendency toward aggregation. The nuclei are often vesicular and spindle-shaped and enlarged in size with a gain in size of 5- to 10-fold than regular. Mitotic figures are frequently observed adjacent to necrosis areas. Moreover, high-grade CHS express and form extensive areas of non-calcified tissue (Figure 1) [17]. 

The grading of CHS by morphologic criteria is sometimes difficult, and thus, immunohistochemical detection of different protein expression patterns can help in distinction. 

High-grade malignant tumors include dedifferentiated and mesenchymal CHS. While the former shows a histological pattern of undifferentiated, small, round, uniform cells and well-differentiated hyaline cartilage areas, the latter is positive for S100 and SRY-box transcription factor 9 (SOX9) staining. This accounts for the conventional subtype as well, but mesenchymal CHS are additionally CD99 and NK2 homeobox 2 (NKX2.2) positive (Table 1) [18,19]. SOX9 is expressed in other types of sarcoma such as osteosarcoma, synovial sarcoma, and others, which makes it unspecific for differentiation purposes, as it functions as the main mediator of chondrogenesis [20,21,22]. To distinguish between osteosarcoma and chondrosarcoma dentine matrix protein (DMP-1), expression can be analyzed, which is uncommon in CHS [23]. Similarly, chondroblastic osteosarcoma will express galectin-1 (GAL-1), which is not the case for conventional CHS [24]. In analogy, expression of FLI-1 is typical for Ewing’s sarcoma, but not for CHS [25]. 

Note that S-100 expression is not reported in enchondroma, but markers for chondroid differentiation such as collagen type II and type X are expressed in both enchondroma and conventional CHS (Figure 2) [26,27,28]. This is not the case for periostin, which is expressed in low-grade CHS but not in enchondroma [28,29]. High levels of lactate-dehydrogenase-A (LDH-A) were found in CHS and are accountable to its treatment resistance [30]. Chondrocyte differentiation is dependent on hedgehog signaling; chondrosarcomas show high expression levels of the hedgehog target genes GLI1 and PTCH1, which upregulates tumor cell proliferation when activated constitutively [31].

In addition, the co-expression of cytoskeletal proteins, such as epithelial membrane antigen (EMA), mucin 1 (MUC1), desmin, myogenin, and myoblast determination protein 1 (MyoD1), is likely [19]. However, Friend leukemia virus integration 1 (FLI-1), smooth muscle actin (SMA), glial fibrillary acidic protein (GFAP), and keratins are entirely negative, and integrase interactor 1 (INI1) expression is retained in mesenchymal CHS [19]. Undifferentiated CHS occur when a portion of conventional low-grade CHS transforms into an aggressive high-grade sarcoma (most commonly undifferentiated pleomorphic sarcoma, osteosarcoma, or other less common high-grade sarcomas such as angiosarcoma, leiomyosarcoma, and rhabdomyosarcoma) [32]. As mentioned previously above, dedifferentiated CHS are most likely negative for S100 in dedifferentiated components, which is an essential difference to the mesenchymal and conventional CHS subtype [33]. In a minor percentage (about 20%) of dedifferentiated CHS, p.Arg132His mutation-specific isocitrate dehydrogenase 1 (IDH1) antibody staining is positive. Both conventional and dedifferentiated components may express mouse double minute 2 homolog (MDM2), programmed cell death receptor ligand 1 (PD-L1), and New York esophageal squamous cell carcinoma 1 (NY-ESO) marker [34]. While aurora kinase, which belongs to the family of serine kinases and is responsible for cell cycle regulation through the control of centriole and microtubule function, is expressed in higher grade CHS, much lower expression rates were detected in low-grade CHS [35,36].

Other histological subtypes of CHS are low-grade clear-cell CHS and periosteal CHS. While the former is characterized by a large proportion of transparent cells with clear, pale cytoplasm in the presence of glycogen vacuoles and distinct cytoplasmic membranes, the periosteal CHS show well-differentiated lobular cartilage areas and moderately cellular cartilage with areas of calcification and endochondral ossification (Figure 3) [37,38].

Extraskeletal myxoid CHS are malignant neoplasms of soft tissue of uncertain differentiation. Histologically, they are characterized by a multinodular architecture and an abundant hypocellular myxoid matrix and interconnecting cords of uniform neoplastic cells with a typical spindle cell differentiation and a high variation in growth pattern. Genetically, NR4A3 gene rearrangement is characteristic for these tumors. Although entitled as CHS, there is no evidence of cartilaginous differentiation [7]. 

## 3. Tumor Microenvironment of CHS

The tumor microenvironment (TME) is of increasing interest to researchers and clinicians in the search for effective targeted therapies. The sarcoma microenvironment is a very complex heterogenous and dynamic milieu, characterized mainly by high interstitial acidosis and high-density immune and genetic heterogeneity [30]. In general, tumor microenvironment is highly vascularized and built by mesenchymal stroma cells that are in close cross-talk with tumor cells [39]. From a clinical perspective, CHS appear to be less vascularized, as intralesional resections are not as bloody as in other entities and, microscopically, there is not a very dense vascular network [40]. The literature investigating mesenchymal stroma cell (MSC) recruitment to the CHS microenvironment is limited. However, in analogy to other types of bone sarcoma, CHS TME is characterized by a heterogenous cell population with an intensive intercellular cross-talk, which enhances tumor growth, progression, and aggressiveness by the secretion of growth factors, cytokines, and extracellular matrix (ECM) deposition [41]. 

MSCs can differentiate toward diverse types of cells, such as myofibroblast-like cells, pericyte-like cells, chondrocytes, adipocytes, osteocytes, and cancer-associated fibroblasts (CAFs) [39]. Furthermore, there is a close cross-talk between MSCs and macrophages, mediating the polarization of macrophages into the M2-like phenotype, also known as tumor-associated macrophages (TAMs) [42]. 

Both oncogenic events that occur during MSC differentiation and the microenvironment that favors malignancy development contribute to tumor progression and strengthen the “seed and soil” theory [43,44]. There is a strict and intensive cross-talk between MSCs and sarcoma cells. As such, local tumor-derived acidosis and tumor-associated osteolysis exert a great impact on MSC stemness [45,46]. Lactate, which is the main driver of tumor acidosis, has a key role in tumor progression; Bonucelli G. et al. demonstrated that MSCs are induced by adjacent (osteo)sarcoma cells to undergo Warburg metabolism, and hence increase lactate production and monocarboxylate transporter 4 (MCT4) expression. In fact, MSC-derived lactate feeds (osteo)sarcoma. Indeed, (osteo)sarcoma cells, through MCT1, import lactate, which drives mitochondrial biogenesis and promotes the migratory skill of (osteo)sarcoma cells [47]. Those mechanisms have not yet been investigated and demonstrated in CHS. 

There was an increase in glucose uptake and a decrease in its oxidation in the sarcoma patients observed, which is indicative of an altered glucose metabolism [46]. An increased glycolytic flux assures several key benefits to cancer cells. For example, mitochondrial oxidation is boosted, leading to a faster ATP production that subsequently results in an advantage for cancer cell growth through elevated energetic resources, and the production of glycolytic intermediates fuels divergent pathways that meet the metabolic demands of proliferating cells [47]. Through different molecular modifications of glycolytic enzyme activity and expression, proliferative cells are able to maintain this high glycolytic flux.

The sarcomas’ TME is also rich in immunosuppressive cytokines, including vascular endothelial growth factor (VEGF). Both VEGF and hypoxia-inducible factor-1 (HIF-1) inhibit the maturation of dendric cells and promote M2 macrophages and regulatory T-cell (T_reg_) migration inside and into the tumor stroma. The increased expression of HIF-2α and negative beclin-1 levels, which mediates autophagy, is predictive for a limited overall survival in CHS, and the inhibition of HIF-2α may lead to an improved malignant signature of CHS [48,49]. Indeed, in sarcomas, increased expression of VEGF and hypoxia correlate with poor prognosis and resistance to chemotherapy [50]. Another characteristic of CHS explaining multidrug resistance in these tumors is the expression of heat shock proteins (HSP) and p-glycoprotein from the MDR-1 gene [51].

The environment of sarcomas is characterized by MSC-induced acidification, promoting tumor growth and epithelial mesenchymal transition and hypoxic conditions [46]. As a result of these sarcomas’ TME features, chemotherapeutics exhibit limited toxicity in this oxygen-deprived and acidic tumor atmosphere.

### 3.1. Immune Microenvironment of CHS

The immune microenvironment of CHS is poorly understood. For future clinical decision-making on whether checkpoint blockade or other types of immunotherapy are effective for the treatment of CHS, a better and more in-depth understanding of the immunoprofile of CHS is needed [52]. Apart from the aforementioned mechanical barriers of the TME, several microenvironmental mechanisms have been proposed to promote chemoresistance in CHS. In this context, the expression of membrane-bound P-glycoprotein, which has been associated with more invasive and higher-grade chondrosarcomas, have to be mentioned [53,54]. The role of increased infiltration of CD163+ M2 macrophages in CHS should be clarified. While in other solid tumors, an increased number of infiltrating CD163+ M2 macrophages correlates with a worse prognosis and a chemotherapy-induced switch from M2 to M1 in osteosarcomas, thus increasing patient survival, it was mentioned that CD163+ infiltration in CHS is associated with a larger tumor mass, but no correlation between CD163+ TAMs and overall patient survival could be confirmed. On the other hand, another recent study found that tumor-associated macrophages were the predominant immune cell type in the immune environment of CHS. Increased levels of CD68+ and CD163+, macrophages correlate with metastatic diseases at diagnosis and there is a limited prognosis for survival [55]. These observations were confirmed recently by Kostine et al. who demonstrated that 41–52% of dedifferentiated chondrosarcomas displayed PD-L1 positivity, which correlated with elevated concentrations of TIL and HLA class I expression [34].

### 3.2. Extracellular Matrix in CHS

The extensive extracellular matrix (ECM) of CHS can be described as proteoglycan rich and is mainly built of structural type II collagen, hydrophobic proteoglycans and hyaluronan. In combination with relatively poor vascularity, high interstitial pressure is generated. These features of CHS ECM further complicate efficient drug delivery to cells for therapeutic intervention. Due to the highly negative charge of proteoglycans, the ECM is extensively hydrated (93% water), with 27% of the dry weight (7% of total weight) [40,56]. Type II collagen is a major protein and is the highest expressed gene in Swarm chondrosarcoma tumors (SRC tumor), which accounts for 50% of the dry weight of the tumor. Aggrecan, which represents more than 90% of the source of sulfate moieties in the tumor, is the major contributing factor [40]. Within this matrix, CHS cells are thoroughly anchored. Growth factors, such as connective tissue growth factor (CTGF), are expressed by human chondrosarcomas and enhance the adhesion of human chondrosarcoma cells through interactions of cells with fibronectin [57,58]. As such, CTGF may moderate distinct stages of metastatic progression, and growth factor expression is upregulated in aggressive tumors [59].

Among many other approaches, the perturbation of the ECM is one possible mechanism to bar tumor growth. Therefore, deregulation of ECM protein expression and secretory trafficking of ECM molecules represent viable therapeutic targets for the treatment of chondrosarcoma [37].

## 4. Genetics of CHS

Heterogeneity is one hallmark of CHS, which are associated with a complex cytogenetic signature [60]. Yet, subtypes of CHS have been shown to have frequent genetic alterations, including mutations in COL2A1, IDH, and the hedgehog signaling pathway [37]. However, so far, no characteristic genomic change has been found. Isocitrate dehydrogenase (IDH)1 and (IDH)2 mutations were identified in about half of CHS analyzed by Amary and colleagues [61].

IDH is a metabolic enzyme that collateralizes the oxidative decarboxylation of isocitrate to alpha-ketoglutarate (a-KG). IDH1 and IDH2 gene mutations have been described in several other malignancies, as well [62,63,64]. Although IDH1/2 mutations were described in cartilaginous neoplasms, including patients with CHS (65% of conventional CHSs and up to 57% of dedifferentiated CHSs), these mutations were not found in the clear-cell and mesenchymal CHS [60,61,65].

The second most frequent mutation in conventional (central and peripheral) and dedifferentiated CHS occurs in the TP53 gene (20–50%) [66]. In multiple studies, the correlation between the overexpression of the TP53 gene or its alteration (loss of heterozygosity on chromosome 17p) and a higher histologic grade of the tumor was observed [67,68].

Further typical gene alterations and mutations were described for the 8q24 region (including c-MYC oncogene) in about one third of high-grade CHS and the 13q14 and 17p13 chromosomal regions for well-differentiated CHS [69].

Furthermore, genes associated with the cell cycle control process, such as cyclin-dependent kinase 4 (CDK4) and MDSM2 are frequently mutated in CHS [68]. Gene fusions are important for the differentiation of sarcoma, but so far are not well-investigated in CHS. One gene fusion recognized in mesenchymal CHS is between hairy/enhancer-of-split related with YRPW motif 1 (HEY1) and nuclear receptor coactivator 2 (NCOA2) genes. This is a deletion or translocation in the 8 chromosome region (q13;q21), which takes part in the process of epithelial–mesenchymal cell transition [70,71].

From an epigenetic perspective, the promoter of the tumor suppressor gene P16INK4a is often hypermethylated in CHS [72]. Similarly, hypermethylation of the promoter region of the tumor suppressor RUNX3 transcription factor leads to a reduced gene expression, increased proliferation, and reduced apoptosis in CHS cells in vitro. Gene expression of RUNX3 is associated with better clinical prognoses, which further demonstrates the importance of DNA methylation and its investigation in CHS research [73].

## 5. Implications for Targeted Therapies

Chondrosarcoma cells are responsive to different soluble factors produced by their microenvironment and activate various tyrosine kinase receptors. Therefore, several tyrosine kinase inhibitors are assessed clinically either alone or in combination (e.g, regorafenib, pazopanib, dasatinib (Bcr-Abl and Src family tyrosine kinase inhibitor), and imatinib (Bcr-Abl, cKIT, RET, NGF-R, PDGFRα/β, ABL1, M-CSFR)).

From multiple studies, it is well understood that mTOR is essential in the control of numerous basic biological cell functions, such as proliferation and migration, and that it acts as a sensor for nutritive elements. One of the most effective and investigated mTOR inhibitors in clinical application is rapamycin (sirolimus), which acts immunosuppressive through mTOR inhibition after its binding to FKBP12. In this context, Bernstein-Molho et al. analyzed the effect of mTOR inhibition by sirolimus combined with cyclophosphamide in a series of 10 recurrent unresectable chondrosarcomas and found a disease control in up to 70% of the cases [74]. Further potential targets for the effective treatment of CHS are angiogenesis inhibitors, as VEGF-A expression has been reported to correlate with tumor grade. Cyclin-dependent kinase inhibitors (CDKis) have demonstrated to be effective for the treatment of breast cancer and are currently US Food and Drug Administration (FDA)-approved to treat this cancer. This may account for CHS as well.

Another potential target is the hedgehog pathway. Misregulation of the hedgehog pathway has been found to cause neoplastic transformations, malignant tumors, and drug resistance of many cancers, including CHS [31]. As mentioned before, inhibition of mutated IDH1 in CHS cells has an adverse prognostic impact on survival, and thus IDH becomes a possible target for CHS therapy. The acidic and lactate-enriched ECM of CHS can be targeted by LDH inhibitors.

Immunotherapies are of growing interest for the treatment of CHS, and PD-1 checkpoint inhibition showed partial success in CHS patients, yet data is still sparse [52].

Other therapy options such as CAR T-cell therapy or targeting miRNA such as miR-100 oder miR-125b have not yet entered into CHS treatment studies [75,76,77,78].

## 6. Conclusions

Chondrosarcomas are a heterogeneous group of rare tumors. Although conventional low-grade chondrosarcomas exhibit locally aggressive behavior, they have a good prognosis with appropriate local surgical therapy. With incising vascularization, conventional and dedifferentiated chondrosarcomas tend to metastasize, often to the lung. In addition, high-grade conventional chondrosarcomas and dedifferentiated chondrosarcomas exhibit remarkable resistance to therapy, including chemotherapy, radiotherapy, as well as targeted approaches. This has been attributed to comparatively slow proliferation, MDR-1 gene expression, relatively poor vascularization, and a dense hyaline ECM. Persistent stem cell formation as well as global epigenetic and genomic changes seem to be implicated in the therapeutic resistance of chondrosarcomas. In addition to the challenge to identify relevant targets for this rare disease, single agent approaches may not be applicable at all. In reality, clinical trials—as listed in Table 2—targeting mutant IDH, HDAC, PI3K/AKT/mTOR and Src signaling as well as VEGF pathways are ongoing and have to be evaluated [49]. Furthermore, immunotherapy has to be considered as a therapeutic option of the future.

## Figures and Tables

**Figure 1 cancers-15-02556-f001:**
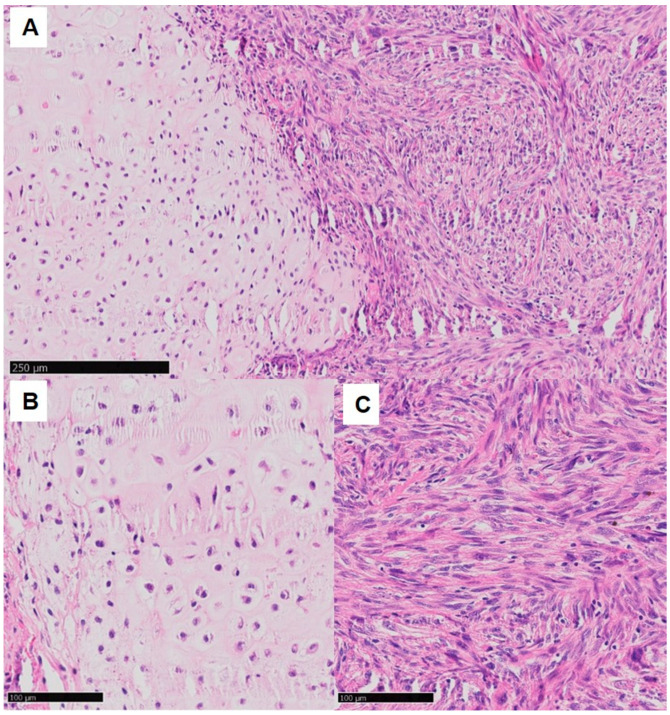
Dedifferentiated chondrosarcoma, 200× (**A**) well-differentiated cartilaginous component; 400× (**B**) low-grade portion of the CHS with atypical chondrocytes and a larger proportion of extracellular matrix; (**C**) high-grade dedifferentiated component with spindle cell appearance.

**Figure 2 cancers-15-02556-f002:**
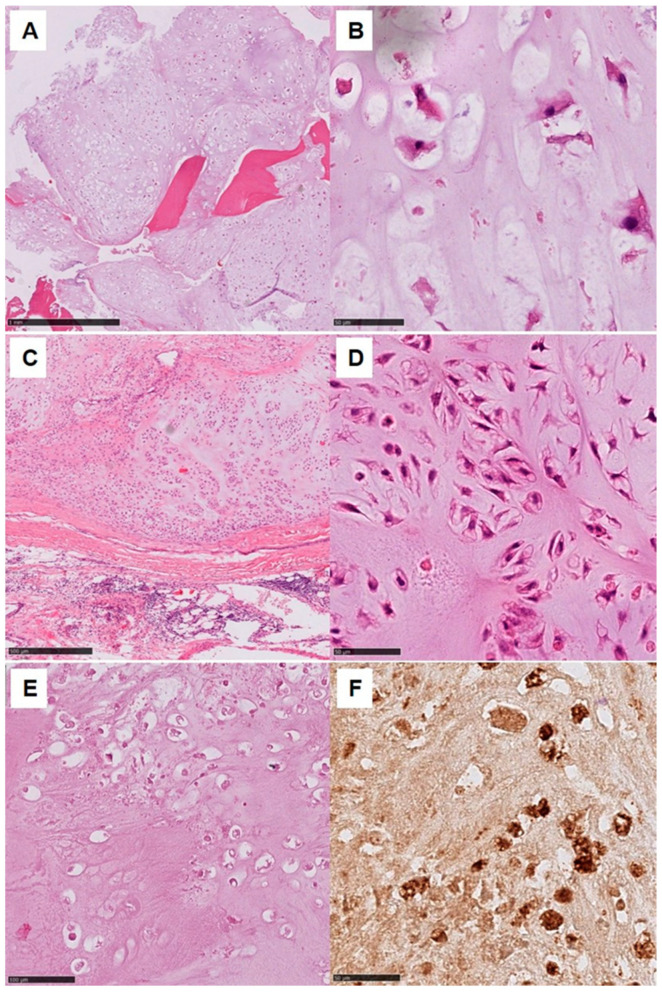
Chondrosarcoma G1-G2 50× (scale bar 1 mm) (**A**); 400× (scale bar 50 µm) (**B**); chondrosarcoma G2 100× (scale bar 500 µm) (**C**); 400× (scale bar 50 µm) (**D**); myxoid degenerated partial necrotic chondrosarcoma 200× (scale bar 100 µm) (**E**); S100 positivity 400× (scale bar 50 µm) (**F**).

**Figure 3 cancers-15-02556-f003:**
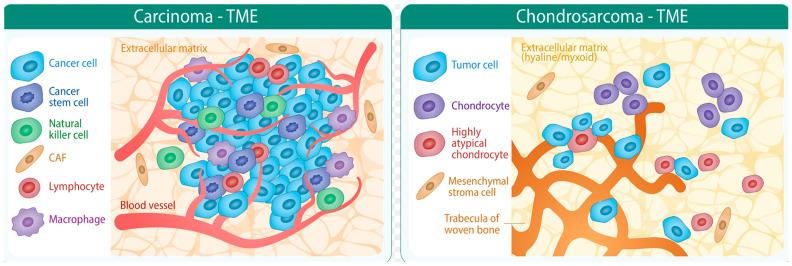
Depicting differences between carcinoma (e.g., G2) and CHS (G1 and G2). Note: Poor vascularization of G1 and G2 CHS under a relatively lower cell density.

**Table 1 cancers-15-02556-t001:** Differential expression of markers among CHS subtypes.

	Conventional CHS	Clear-Cell CHS	Mesenchymal CHS	High-Grade CHS
S-100	+	+	+	−
NY-ESO	+	−	−	+
Ezrin	−	−	+	+
IDH	+	−	−	+
Other	D2-40, osteonectin, MDM2, Cox-2	Runx2, MYF4, keratine	CD99, MYOD1, NKX2.2, desmin, vimentin	CD44, MDM2, Runx2, PD-L1

**Table 2 cancers-15-02556-t002:** Overview of the current approaches for targeting CHS. CS, clinical study.

Target Group	Mechanism	Study Type	Promising	Active Agent	Studies
Angiogenesis	Endothelial cell activity	CS	+	Plasminogen-related protein B	[79]
	VEGF inhibition	CS		VEGF-AB	[80]
	VEGF pathway inhibition VEGF antibody	CS	+/−	Pazopanib, ramucirumab	[81,82]
	VEGF pathway inhibition	CS	+	Regorafenib	[83]
Cyclin-dependent kinase	CDK-4 inhibition	In vitro		Palbociclib	[84]
Hedgehog	Deregulation of the Hedgehog pathway	In vitro		HPI-4	[85]
IDH	Mutant IDH inhibition	CS	+	Ivosidenib (AG-120)	[86]
	Mutant IDH1 inhibition	In vitro	+	AGI-5198	[87]
	Mutant IDH1 inhibition	In vitro, in vivo model	+	DS-1001b	[88]
Tyrosine kinase	Tyrosine kinase inhibition	CS	−	Dasatinib	[89]
	Tyrosine kinase inhibition	CS	−	Imatinib	[90]
mTOR	mTOR pathway dysregulation	CS	+	Sirolimus	[74]
	mTOR pathway dysregulation	In vivo model	+	Everolimus	[91]
Immune checkpoints	PD-L1 inhibition	CS	+/−	Pembrolizumab	[92]
	Anti-PD1 therapy	CS	+/−	Nivolumab	[93]
LDH	LDH inhibition	In vivo model		NCI-737	[94]
ECM	MMP inhibition	In vivo model		QA-Dox	[95]

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
