# Peer review of "Molecular In-Depth Characterization of Chondrosarcoma for Current and Future Targeted Therapies"

_cancers, 2023, doi:10.3390/cancers15092556_

Round 1

Reviewer 1 Report

The authors need to double check this short paragraph for accuracy especially the last two points where it is claimed that MSCs differentiate to TAMs and dedifferentiate to sarcoma tumor cells:

lines 171-173

"MSCs can differentiate towards diverse types of cell such as myofibroblast-like cells, pericyte-like cells, chondrocytes, adipocytes, osteocytes, cancer associated fibroblasts (CAFs) and tumor associated macrophages (TAMs) and dedifferentiate into sarcoma tumor cells"

minor edit

Future advances in CHS therapy are depend on further characterization of CHS especially the tumor immune microenvironment for better and targeted therapies

please omit "are"

Author Response

Thank you very much for your comment.

We changed this paragraph into:

"

MSCs can differentiate towards diverse types of cell such as myofibroblast-like cells, pericyte-like cells, chondrocytes, adipocytes, osteocytes and cancer associated fibroblasts (CAFs). [38] Furthermore, MSC there is a close cross-talk between MSCs and macrophages, mediating polarization of macrophages into the M2-like phenotype also known as tumor associated macrophages (TAMs).[41]"

Future advances in CHS therapy are depend on further characterization of CHS especially the tumor immune microenvironment for better and targeted therapies

This sentence was changed into:

"Future advances in CHS therapy depend on further characterization of CHS especially the tumor immune microenvironment for better and targeted therapies"

Reviewer 2 Report

The present review "Molecular in-depth characterization of Chondrosarcoma for current and future targeted therapies" submitted by Sebastian G. Walter et al. is a valuable submission however some minor improvements are recommended. The manuscript can be considered for publication after revising the following points:

Minor concern:

1- Why were two values reported for the 5-year survival rates instead of one? What does the second value represent? It is not apparent to the reader (e.g. line 60, 61).

2- Figure 1 was misnamed and it shows three sections, but a part is missing in the legend.

3- The authors should double-check if the magnification in the legend for Figure 2F is correct. Should it be 400x (line 124)?

3- The authors should spell out each abbreviation at its first mention (line 236: SRC tumor; line 278: CS (or CHS?))

4- At the conclusion, it is mentioned for the first time that conventional and dedifferentiated chondrosarcomas tend to metastasize, often to the lung (line 314). This information should also be included in the introduction and supported with a citation."

5- The authors should also include citations for the clinical trials (line 322).

6- There is an extra doi in reference 39.

Author Response

Thank your very much for your comments.

Below we tried to answer your valuable remarks:

  1. We changed the paragraph into: "Survival rate of CHS depends on subtypes, with good 5-year survival rates in the periosteal subtype (68.1%) and conventional CHS (68.4%). The 1 and 5-year survival rates in clear cell (88.7%, 62.3%), myxoid (86.2%, 49.8%), and mesenchymal (76.1%, 37.6%) subtype are somewhat lower but still better than in dedifferentiated CHS with reported 5-year survival rates between 11 %and 24%.[12]"
  2. The description of the figure was changed into: "

    Figure 1: Dedifferentiated chondrosarcoma 200x (A) well differentiated cartilaginous component;  400x (B) low grade portion of the CHS with atypical chondrocytes and a larger proportion of extracellular matrix, (C) High grade dedifferentiated component with spindle cell-appearance"

  3. Thank you very much for your comment. Indeed it is 400x magnification. The legend was changed accordingly.
    1. According to your remark regarding abbreviations CS the sentence in ll. 286-287 was changed into: "Gene expression of RUNX3 is associated with better clinical prognoses, which further demonstrates the importance of DNA methylation and its investigation in CHS research"
  4. Thank you for this remark. The following sentence was added to the introduction: "In clinical practice, CHS are still classified into three grades based on their histology, ranging from well-differentiated, low-cellularity, exceptionally metastatic grade I CHS to poorly differentiated and highly cellular grade III CHS with high risk for pulmonary metastasis. [6–8]"
  5. The sentence was changed to: ". Actually, clinical trials -as listed in Table 2- targeting mutant IDH, HDAC, PI3K/AKT/mTOR and Src signaling as well as VEGF pathways are ongoing and have to be evaluated. "
  6. Thank you for this remark. The double inserted doi was removed.

Reviewer 3 Report

The manuscript of Walter et al. is a review article, aiming to better interpret the molecular characterization of chondrosarcoma in order to improve its treatment, in particular when surgery is not possible.

The authors proposed first to explain the different grades and subtypes of chondrosarcoma according to their location in relation to the bone.

Then, the morphology and histological analysis of subtypes are described according to the grade.

A deep description of the role of the microenvironment is proposed, including the role of the different cellular subtypes, the immune microenvironment and the extracellular matrix.

The authors tried then to give a general view of the “genetics” of chondrosarcoma and the related targeted therapies.

The presented manuscript is reasonably well written.

I have some comments that could improve the overall quality of the manuscript.

1) Please harmonize the way to cite references: add the reference within the sentence and before the “.”

Examples: P2 L39-40:

Change : Primary malignant bone tumors are a rare entity and thus account for less than 1% of cancers.[1]

To : Primary malignant bone tumors are a rare entity and thus account for less than 1% of cancers [1].

All references should be checked

2) P3: it should be “figure 1” ? in the legend, where is “the part “C” ?

3) Regarding the genetics of chondrosarcoma and the possible targeted therapy, the authors should include the following studies in link with IDH and HIF:

10.1038/s41467-020-18817-7

10.1038/s41388-019-0929-9

Considering the overall quality of this study; I recommend this manuscript for publication after these changes

Author Response

Thank you very much for your comments.

  1. All references were checked and changed according to your proposal.
  2. The legend of the figure was changed and all parts are addressed now.
  3. Both studies are now cited within the manuscript

Reviewer 4 Report

I read this manuscript with interest.  The authors need to check the following contents in the manuscript:

1. In page 2 line 95, it seems there should have a figure 1 in the manuscript, however, I cannot find it in the manuscript.

2. In Figure 2, the authors miss the explanation for C, please clarify what was C in your figure 2.

Author Response

Thank you very much for your comments.

  1. The legend of figure 1 (formerly figure 2) was changed accordingly
  2. The legend now accounts for all parts of the figure, including part C